# PHAGE Study: Effects of Supplemental Bacteriophage Intake on Inflammation and Gut Microbiota in Healthy Adults

**DOI:** 10.3390/nu11030666

**Published:** 2019-03-20

**Authors:** Hallie P. Febvre, Sangeeta Rao, Melinda Gindin, Natalie D. M. Goodwin, Elijah Finer, Jorge S. Vivanco, Shen Lu, Daniel K. Manter, Taylor C. Wallace, Tiffany L. Weir

**Affiliations:** 1Department of Food Science and Human Nutrition, Colorado State University, Fort Collins, CO 80523, USA; Hallie.Febvre@colostate.edu (H.P.F.); melinda@stillwater.life (M.G.); natalie.goodwin@colostate.edu (N.D.M.G.); elijah.finer@colostate.edu (E.F.); shen.lu2017@hotmail.com (S.L.); 2Department of Clinical Sciences, Colorado State University, Fort Collins, CO 80523, USA; sangeeta.rao@colostate.edu; 3Polaris Expeditionary Learning School, Fort Collins, CO 80525, USA; 91929@psdschools.org; 4Soil Management and Sugarbeet Research, ARS, USDA, Fort Collins, CO 80523, USA; daniel.manter@ars.usda.gov; 5Think Healthy Group, Inc., Washington, DC 20001, USA; 6Department of Nutrition and Food Studies, George Mason University, Fairfax, VA 220030, USA

**Keywords:** bacteriophage, cytokines, gut microbiota, gastrointestinal, inflammation, short-chain fatty acid

## Abstract

The gut microbiota is increasingly recognized as an important modulator of human health. As such, there is a growing need to identify effective means of selectively modifying gut microbial communities. Bacteriophages, which were briefly utilized as clinical antimicrobials in the early 20th century, present an opportunity to selectively reduce populations of undesirable microorganisms. However, whether intentional consumption of specific bacteriophages affects overall gut ecology is not yet known. Using a commercial cocktail of *Escherichia coli*-targeting bacteriophages, we examined their effects on gut microbiota and markers of intestinal and systemic inflammation in a healthy human population. In a double-blinded, placebo-controlled crossover trial, normal to overweight adults consumed bacteriophages for 28 days. Stool and blood samples were collected and used to examine inflammatory markers, lipid metabolism, and gut microbiota. Reductions in fecal *E. coli* loads were observed with phage consumption. However, there were no significant changes to alpha and beta diversity parameters, suggesting that consumed phages did not globally disrupt the microbiota. However, specific populations were altered in response to treatment, including increases in members of the butyrate-producing genera *Eubacterium* and a decreased proportion of taxa most closely related to *Clostridium perfringens*. Short-chain fatty acid production, inflammatory markers, and lipid metabolism were largely unaltered, but there was a small but significant decrease in circulating interleukin-4 (Il-4). Together, these data demonstrate the potential of bacteriophages to selectively reduce target organisms without global disruption of the gut community.

## 1. Introduction

Intestinal health and the gut microbiota have increasingly been linked to various chronic health outcomes. Imbalances in the gut microbiota resulting from poor diet, stress, antibiotic use, and other lifestyle and environmental factors are associated with the development of intestinal inflammation and bowel irregularities [1,2]. Several autoimmune and metabolic conditions and even mental health may also be rooted in the gut and influenced by its microbial residents [2]. As a result, there is a growing interest in identifying dietary supplements that favorably modulate gut microbial populations. According to Transparency Market Research, the U.S. market for probiotics, prebiotics, and other digestive aids currently exceeds $68.8 billion and is expected to reach $83.5 billion by 2022 [3]. Thus, there exists a critical need to identify safe and effective methods of manipulating gut microbiota to promote overall health and well-being.

Bacteriophages (or phages) are among the candidates being explored as potential microbial modifiers in promoting intestinal health [4]. These ubiquitous, bacteria-targeting viruses exhibit a high degree of host specificity, suggesting utility for selectively reducing pathogenic or pro-inflammatory bacteria in the microbial milieu. The antibacterial activity of phages was first observed in the waters of the Ganges and Jumna rivers in India in 1896 [5]. In 1917, Felix d’Herelle demonstrated their clinical relevance by isolating and applying phages to treat numerous bacterial infections [6]. However, despite promising early results, the concept of phage therapy lost momentum with the introduction of broad spectrum antibiotics, which allowed the treatment of bacterial diseases without the need to identify a specific causal organism [7]. The host specificity of phages, which has limited their widespread application as clinical antimicrobials, may be advantageous when considering their use as microbiota-modulating dietary supplements. While antibiotics can cause or exacerbate microbiota imbalances or dysbiosis [8], phages offer the opportunity to subtly and selectively modify the gut microbiota. Several bacteriophages are “Generally Recognized As Safe” (GRAS) for human consumption by the U.S. Food and Drug Administration (FDA). Recently, we demonstrated that oral ingestion of a bacteriophage cocktail in dietary supplement form was safe and tolerable in a healthy adult population [9].

The objective of the current study was to determine how daily consumption of supplemental *Escherichia coli*-targeting phages (commercially sold as PreforPro^®^) influences the gut microbiota of healthy adults with self-reported gastrointestinal distress. In addition, we determined the effects on microbial production of short-chain fatty acids (SCFAs), as well as explored whether phage consumption alters lipid metabolism and parameters of local and systemic inflammation. Here, we report that 28 days of phage consumption did not substantially alter the global gut microbiota profiles of most individuals, but did reduce populations of the target bacteria, *E. coli*, as well as modify a number of individual bacterial species, including an increase in amplicon sequence variants (ASVs) mapping to *Eubacterium spp*., which is one of the most abundant genera in the healthy human gut [10]. We also saw a reduction in the circulating pro-inflammatory cytokine interleukin 4 (Il-4), which has been associated with autoimmune and allergic responses in human populations [11]. These data highlight the potential of bacteriophages for selective modification of targeted microbial species without inducing dysbiosis.

## 2. Materials and Methods 

### 2.1. Participant Characteristics

Forty-three healthy adults aged 18–65 with self-reported gastrointestinal issues were enrolled in the study. Eligibility was determined by a telephone or in-person screening prior to obtaining informed consent. Exclusion criteria included (a) a previous diagnosis of gastrointestinal or metabolic conditions, cancer, liver, or kidney diseases; (b) pregnancy or breastfeeding;(c) smoking;(d) use of antibiotics in the last 2 months; and (e) current medication or dietary supplement use that may impact gut microbiota. Participants were asked to maintain their regular diet and exercise habits throughout the study, refrain from supplemental prebiotics or probiotics, and limit alcohol consumption to 1 drink per day or no more than 7 drinks per week. Additional study criteria and participant demographics are reported in Gindin et al. [9]. Of the total enrolled participants (*n* = 43), 36 completed at least one arm of the study and 32 of those completed the entire study. The data shown here represent analysis from these 36 participants.

### 2.2. Study Design 

The study was conducted as a randomized, double-blind, placebo-controlled crossover intervention trial, with two 28 day intervention periods and a washout period of at least two weeks between treatments. The treatments consisted of 4 supplemental bacteriophage strains (LH01-*Myoviridae*, LL5-*Siphoviridae*, T4D-*Myoviridae*, and LL12-*Myoviridae*) at a titer of 10^6^ phages per dose included in PreforPro^®^ commercial capsules prepared by Deerland Enzymes (Kennesaw, GA, USA). The placebo was rice maltodextrin and coconut triglycerides, which were also the carrier materials used in the treatment capsules. Participants were asked to consume one 15 mg capsule per day during the treatment and placebo periods. The study was conducted at the Human Performance Clinical Research Laboratory (HPCRL) at Colorado State University. At each clinic visit (*t* = 0 and *t* = 28 days for each intervention period, 4 visits in total),participants provided a fresh stool sample that had been collected at home and returned to the clinic within 24 h of their visit. A fasted blood sample was also collected. Collected stool and plasma samples were stored at −80 °C prior to analysis. A detailed study protocol has been previously published [9]. The study protocol was approved by the Institutional Review Board (IRB) for Human Subjects Research at Colorado State University, CSU protocol #16-6666HH, and all participants provided written informed consent prior to beginning the study. The study is also registered at clinicaltrials.gov as NCT03269617 [9].

### 2.3. DNA Extraction and Sequencing

Collected stool samples were thawed and subsampled with sterile cotton swabs. Fecal DNA was then extracted from the swabs using the FastDNA^®^ KIT (MP Biomedical; Santa Ana, CA, USA) following modified manufacturer’s instructions that included optimization with additional wash steps. Sequencing libraries were constructed by PCR amplification of the V4 region of the *16s* rRNA gene using primers 515F and 806R following the protocol for the Earth Microbiome Project [12]. Amplicons were purified using AxyPrep Mag PCR clean-up beads (Axygen; Corning, NY, USA), and amplicons were quantified with a Quanti-iT PicoGreen dsDNA Assay Kit (Invitrogen; Eugene, OR, USA) and pooled in equimolar ratios prior to sequencing at the Colorado State University Genomics Core facility using a 2 × 250 MiSeq flow cell (Illumina, San Diego, CA, USA).

### 2.4. Microbiota Analysis 

Paired-end sequence reads were concatenated, and all combined 16s sequences were filtered, trimmed, and processed with the DADA2 [13] implementation included in the open source bioinformatics tool myPhyloDB version 1.2.1 [14]. Briefly, all primers were removed from each sequence using the open source Python program Cutadapt [15], and amplicon sequence variants (ASVs) were inferred using the default pipeline in DADA2. Each sequence variant identified in DADA2 was classified to the closest reference sequence contained in the GreenGenes reference database (Vers. 13_5_99) using the usearch_global option (minimum identity of 97%) contained in the open source program VSEARCH [16].Analysis of covariance (ANCOVA), DiffAbund, and principle coordinates analysis (PCoA) analyses were conducted in myPhyloDB, and MicrobiomeAnalyst [17] was used to calculate alpha diversity scores. Data were filtered to remove unassigned reads and phylotypes with ≥25% zeros. The raw sequencing data are available upon request, and links to the metadata and.biom files are included with the Appendix A.

### 2.5. Short-Chain Fatty Acids 

Short chain fatty acids (SCFA) were extracted from frozen fecal samples and analyzed as previously described [18]. Briefly, fecal aliquots were extracted in acidified water (pH 2.5) containing an internal standard of 5mM of ethylbutyric acid. Suspended samples were homogenized and sonicated, followed by centrifugation to remove particulate matter. Supernatant was analyzed on a Gas Chromatograph with Flame Ionization Detection (GC-FID; Agilent 6890 Plus GC Series, Agilent 7683 Injector series, GC Column: TG-WAXMS A 30mx 0.25mm × 0.25μm). The run program was as follows: Initial temp = 100 °C for 1 min, max temp = 300 °C, equilibration time = 1 min with a ramp rate of 8.0 for 1 min to a final temp of 180 °C, followed by a ramp rate of 20.0 to a final temp of 200 °C for 5 min. Post temperature was 50 °C. The front inlet was split (10:1), 240 °C, 16.57psi. Peak areas were normalized to the internal standard (5 mM ethyl butyric acid, Retention Time (RT) = 9.2) and quantified using standard curves (acetic acid, RT = 5.5; propionate, RT = 6.7; butyrate, RT = 7.7) from dilutions of commercial stocks.

### 2.6. Fecal Triglycerides

Fecal triglycerides were assessed using the Triglycerides Assay Kit (Cayman Chemicals, Ann Arbor, MI). Briefly, 75 mg of homogenized fecal sample was suspended in 1xNP40 reagent containing protease inhibitors. Samples were centrifuged at 4 °C for 10 min at 10,000 rpm. Supernatant was diluted 1:5 with 1xNP40, and absorbance at 530–550nm was measured after incubation for 15 min at room temperature. Triglyceride quantity was determined by fitting to standard curves.

### 2.7. Local Inflammation and Immune Responses

Fecal secretory immunoglobulin A (sIgA) and fecal calprotectin were analyzed using the Human Secretory IgA ELISA Assay Kit and Calprotectin ELISA Assay kits (Eagle Biosciences, Amherst, NH, USA).

### 2.8. Systemic Inflammation 

Systemic inflammation was assessed by measuring plasma levels of C-reactive protein (CRP), as well as several chemokines and cytokines. CRP levels were assessed using Human hsCRP ELISA kits (BioVender LLC., Asheville, NC, USA) according to the manufacturer’s instructions. In addition, 13 different chemokines and cytokines, which included GM-CSF, IFNγ, IL-1α, IL-2, IL-4, IL-5, IL-6, IL-7, IL-8, IL-10, IL-12 (p70), IL-13, and TNF-α, were measured using the Milliplex MAP Human High Sensitivity T Cell panel (Millipore Sigma, Burlington, MA, USA). All samples were processed according to the manufacturers’ protocols and analyzed on a Luminex instrument (LX200; Millipore, Austin, TX, USA).

### 2.9. Plasma Lipids 

Lipid panels (total cholesterol (TC), high-density lipoprotein cholesterol (HDL), triglycerides (TRIGs), nonhigh-density lipoprotein cholesterol (nHDL), total cholesterol/HDL ratio (TC/HDL), low-density lipoprotein cholesterol (LDL), and very low-density lipoprotein cholesterol (VLDL)) were assayed within one hour of blood collection using a Piccolo Xpress Chemistry Blood Analyzer (Abaxis, Union City, CA, USA).

### 2.10. Statistical Analysis

To evaluate the effects of the starting sequence, differences between baseline levels were assessed for each sequence (A-B, B-A). Continuous data were tested for normality prior to performing linear regression analysis. A linear mixed model approach, controlling for sequence and repeated measures, was used to compare treatment effects within each time point and between time points within a treatment group. A *p*-value of 0.05 was used to assess statistical significance. Prior to statistical analysis, microbiota data were normalized using Laplace smoothing followed [19] by subsampling with replacement (rarefaction (keep) command) in MyPhyloDB [14]. Data were rarefied to 31,037 sequence reads using 100 iterations. Amplicon sequence variants (ASVs) that were present in less than 25% of the total samples were excluded from analyses. An analysis of covariance (ANCOVA) model was used to assess taxonomic differences across treatment groups, and a genewise negative binomial generalized linear model (GLM) (EdgeR [20]) was used to determine differential distribution of taxa between treatments. Measures of alpha (CHAO1 estimates, Shannon and Simpson diversity indices) and beta diversity (Bray–Curtis distances) were statistically analyzed using nonparametric Kruskal–Wallis tests. 

## 3. Results

Study compliance and the safety and tolerability of phage consumption have previously been reported [9]. Here, we present the results of the *16s rRNA* sequencing and targeted metabolite analysis from stool samples as well as plasma lipids and markers of inflammation.

### 3.1. Gut Microbiota and Metabolite Analysis

Bacterial sequences were classified to seven phyla, with the majority being represented by Firmicutes, followed by Bacteroidetes and minor components including Actinobacteria, Proteobacteria, and Verrucomicrobia. There were no significant differences in bacterial taxa between treatment groups and time points (Figure 1A and Appendix A). In addition, richness estimates (CHAO1) and α-diversity, calculated as Shannon and Simpson’s indices, did not vary across groups (Appendix A). PCoA visualization of Bray–Curtis distances did not show any significant clustering between treatment groups or time points (Figure 1B).

Because the supplemented phage cocktail specifically targeted *E. coli,* we identified ASVs in each sample that mapped to *E. coli* (gg_111717). Only 21 total participants had detectable levels of *E. coli* prior to starting the treatment period. Baseline levels varied significantly among participants, ranging from 0.01%–3.2% of total reads. The response rate was ~71%, with 15 of the 21 participants that had detectable levels of *E. coli* prior to starting the treatment period showing reduced or undetectable levels after treatment (responder subpopulation), while only 47% of the individuals with detectable levels of *E. coli* prior to the placebo period showed reduction after 28 days. Overall, *E. coli* levels were significantly reduced after phage treatment (*p* = 0.03 for the total study population and *p* = 0.02 when only including participants with baseline *E. coli*, but not after placebo (*p* = 0.85 for the total study population and *p* = 0.78 for participants with baseline *E. coli*)) (Figure 2). On average, the number of *E. coli* reads were reduced by ~40% after treatment and by only 14% after the placebo, although there was a great deal of variability between participants.

Several microbial taxa were significantly correlated with fecal levels of *E. coli*, regardless of intervention period. *Oxalobacter formigenes* and ASVs assigned to the Lachnospiraceae family were negatively correlated with *E. coli* populations. Conversely, several ASVs identified as belonging to the Ruminococcaceae family were positively correlated with *E. coli*, as was *Desulfovibrio* (Figure 3). 

There was only one bacterial ASV that increased significantly (*q* < 0.10) after phage consumption in the responder subpopulation (those who had baseline *E. coli* and saw reductions with treatment), *Bifidobacterium bifidum*, which was ~5.5-fold higher compared to baseline measures (mean *t* = 0 = 0.13; mean *t* = 28 = 1.1; log fold change (logFC) = 2.7, Counts Per Million (CPM) = 8.1, likelihood ratio = 6.7; False Discover Rate (FDR) = 0.06). There were also several treatment-associated taxa changes noted in the total study population, which included 36 participants that completed at least one arm of the study, regardless of whether *E. coli* was detected in baseline samples. Most notably, three ASVs representing potentially pathogenic or inflammation-associated taxa were reduced relative to the placebo, including an ASV mapping to *Clostridium perfringens*, a food-borne pathogen and minor component of the commensal flora [21], which was reduced by about 75%. In addition, two ASVs representing species of *Eubacterium* were increased by 4–5-fold after phage consumption compared to the placebo (Figure 4).

To assess functional changes in the microbiota, we also assayed fecal short-chain fatty acid (SCFA) concentrations. Acetate was the most abundant SCFA, averaging 10–20 mM/gram of stool across all treatments and timepoints (Appendix A). Propionate and butyrate were detected in levels of approximately 2–4 mM/gram of stool (Appendix A). There were no significant differences in any of the SCFAs detected across timepoints or between treatment periods, although there was a trend toward decreased acetate from the baseline level within the placebo period (*p* = 0.06; *Confidence Interval* (*CI*) = −0.09 to 3.15).

### 3.2. Stool and Plasma Lipid Profiles

Since the microbiota also plays a role in lipid absorption [22,23], we measured total triglyceride levels in stool. Although there was significant interindividual variability, there were no significant differences in fecal triglycerides across timepoints or between treatment periods (Table 1), although there was a significant period effect noted for participants starting on the placebo and transitioning to the treatment group (*p* = 0.03; CI = −8.93 to −0.49). Likewise, there were no significant changes across time periods or between treatments for plasma lipids associated with phage consumption. However, there was a statistically significant change in total cholesterol to HDL ratio (TC/H) between the baseline (*t* = day 0) and day 28 of the placebo period (*p* = 0.045; CI = 0.00 to 0.18), possibly driven by a trend toward reduced HDL cholesterol during this period (*p* = 0.08; CI = −3.87 to −1.75) (Table 1). However, these statistical differences are unlikely to have any clinical or biological relevance.

### 3.3. Immunological and Inflammatory Markers

We examined several stool and blood markers indicative of inflammatory state and immunological activity. In stool, we measured calprotectin and secretory immunoglobin A (sIgA). Calprotectin was below the detection limits of our assay for the majority of samples tested (data not shown): sIgA was detectable, and the majority of samples fell within clinically normal ranges (510–2040 µg/mL). Therewere no significant differences in this parameter across timepoints or between treatment periods (Appendix A), although there was large variability between individuals and even between timepoints for select individuals. In plasma, we analyzed C-reactive protein using a high-sensitivity ELISA assay (hsCRP) as well as a panel of 13 human T-cell-derived cytokines. There were no significant responses across timepoints or between treatments in these parameters, with one exception. Interleukin-4 (Il-4) was significantly reduced from baseline after 28 days of phage consumption (*p* = 0.002; CI = −15.63 to −3.67) (Table 2).

## 4. Discussion

Bacteriophages offer a novel and selective means of modifying the gut microbiota, thereby influencing the intestinal environment without causing global perturbations that can lead to microbial dysbiosis. In the current study, we confirmed that phage treatment was not associated with global perturbations in the gut microbiota, as evidenced by a lack of differences in community descriptors such as richness and α-diversity between treatments or over time. Furthermore, no clustering of groups was apparent on a PCoA plot of Bray–Curtis distances. On the contrary, disruption of microbial communities by antibiotics and other pharmaceutical treatments can predispose individuals to dysbiosis and create ecological niches where pathogens can establish a foothold, as is commonly seen with *Clostridium difficile* infections [8,24]. Thus, phages represent a novel means of selectively modifying the microbiota without causing global disruptions to community structure.

Although no global changes to the microbiota were apparent with phage treatment, there were a few significant alterations in certain members of the microbial community. Importantly, *E. coli*, which is the target host for the consumed phage consortium, was significantly reduced at the end of the treatment period. Populations of several bacterial taxa were also positively or negatively correlated with levels of *E. coli*. While it is difficult to discern whether there were specific ecological interactions driving these associations, both *E. coli* and sulfate-reducing bacteria (SRB), such as *Desulfovibrio*, have been associated with higher clinical activity indices and sigmoidoscopy scores in rectal biopsies of patients with Inflammatory Bowel Disease (IBD) [25,26,27].One study also reported higher *Clostridium perfringens* and lower *Eubacterium* spp. associated with disease severity [25]. Interestingly, consumption of bacteriophages in the current study resulted in a more than 4-fold reduction of *C. perfringens* as well as 4–5-fold increases in two ASVs mapping to taxa in the genus *Eubacterium*. *Eubacterium* reductions have also been associated with several inflammatory conditions in the gastrointestinal tract [28,29]. *Eubacterium* spp. are butyrate producers, and as such may be important in stimulating enterocyte turnover and maintaining tight barrier junctions. However, we did not observe an overall increase in stool butyrate with the bacteriophage treatment. This could have been due to a high level of interindividual variation in SCFA production masking responses at the population level. It may also have been due to insufficient levels of fiber intake, as this is the substrate for SCFA production. Finally, another interesting observation was the positive correlation between the sulfate-reducing bacteria (SRB) *Desulfovibrio* and *E. coli* ASVs across the population. Although the physiological role of SRB in the gastrointestinal tract is still not well understood, it is commonly accepted that production of hydrogen sulfide could have direct inflammatory, cytotoxic, and genotoxic effects in the gut that could reduce epithelial barrier integrity [30]. These data suggest that further investigation of bacteriophages is warranted to determine the therapeutic value of treating individuals with inflammatory bowel conditions.

Another observation associated with phage consumption was a reduction of circulating Il-4. This cytokine is released during Th2 responses, which are associated with the promotion of IgE and eosinophilic responses to atopy [31]. Although the mechanisms relating Il-4 reduction to phage consumption are unclear, some studies have demonstrated that bacterial lipopolysaccharide (LPS) induces Il-4 production via a MyD88 and TRAM-dependent pathway [32]. Although we do not have direct evidence of reduced circulating LPS, we have previously reported lower aspartate aminotransferase (AST) and alkaline phosphatase (ALP) in whole blood samples collected after treatment compared to the placebo control [9]. In a rodent model, AST and ALP increased after exposure to LPS [33]. Circulating LPS is associated with systemic inflammation and increased cytokine release [34]. Therefore, it is plausible that the phage treatment resulted in lower circulating LPS, which may drive reductions in Il-4. These data suggest that future experiments are justified to further explore mechanistic links between Il-4 and phage consumption as well as to examine the effect of the bacteriophage cocktail in a human population with atopic dermatitis and other allergic atopies. 

A major strength of the current study was its crossover design, in which each individual served as his or her own control. This was advantageous given the inter-person variability of the microbiota, as well as individual responses to a stimulus. An additional strength of this study design was the double blinding, thus minimizing participant and researcher bias. Despite these strengths, there were several shortcomings that limited our ability to interpret the data. For example, the phage cocktail prescribed has been shown in animal models to reduce target *E. coli* populations and stimulate the growth of probiotic species [35]. However, the variability in detectable baseline levels of both *E. coli* and these probiotic species in our sample pool made it difficult to replicate these findings in a human population. However, this product is currently marketed as a dietary supplement, and thus it is important to determine how it impacts the global microbiota in a healthy population. While we did see a phage-associated increase in taxa mapping to *Bifidobacterium bifidum* in a subpopulation of responders, examining the influence of the co-administration of phages with particular probiotic species may be necessary to establish whether they enhance probiotic survival and efficacy in the gastrointestinal tract. Another limitation was the lack of a diet and physical activity assessment, as these parameters may influence microbiota composition. While participants were asked to remain on their typical diet and exercise regime and were excluded from the study if they indicated that they were actively trying to lose weight, tracking these metrics would have been an additional measure to assess compliance. Thus, although some of the observed changes in specific bacterial populations may have been in response to changes in these external factors, the study was designed to minimize these effects.

In conclusion, bacteriophage consumption caused minimal disruption to the gut microbiota, but did elicit minor changes that may be viewed as beneficial overall. Specifically, the reduction of *E. coli*, decreased proportions of potentially pro-inflammatory bacteria, and increases in fermentative taxa that are capable of butyrate production suggest a shift toward a healthier gut environment. As gut dysbiosis continues to be associated with human disease and the medical community is confronted with anti-microbial “superbugs”, bacteriophages offer an additional resource to combat these issues. This study merely broaches the edge of this potential, and along with our previously published data [9], confirms the safety and tolerability of phage consumption in a human population. These data support further studies on the microbiota modulatory potential of bacteriophages for use as a dietary supplement and possibly as a therapeutic agent in clinical populations of intestinal inflammation. 

## Figures and Tables

**Figure 1 nutrients-11-00666-f001:**
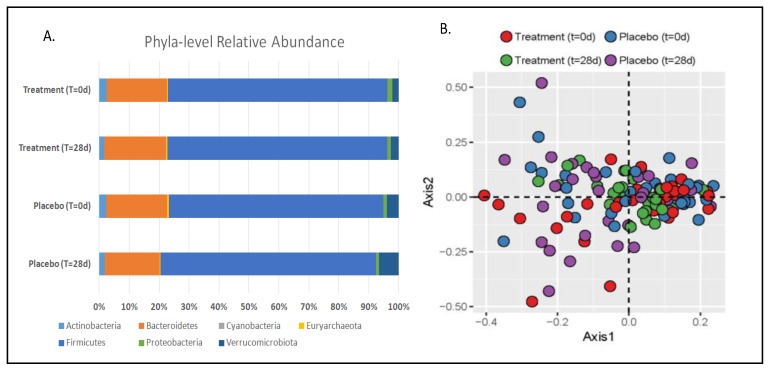
(**A**) Relative abundance of bacterial phyla detected in stool samples at baseline and after 28days for both phage (treatment) and placebo study periods. No significant differences were detected by analysis of covariance (ANCOVA) at *p* < 0.05. (**B**) Principle coordinates analysis (PCoA) with nonmetric dimensional scaling of species-level Bray–Curtis distances. Stress = 0.191; perMANOVA (1000 permutations) *Pr* (>*F*) = 0.996.

**Figure 2 nutrients-11-00666-f002:**
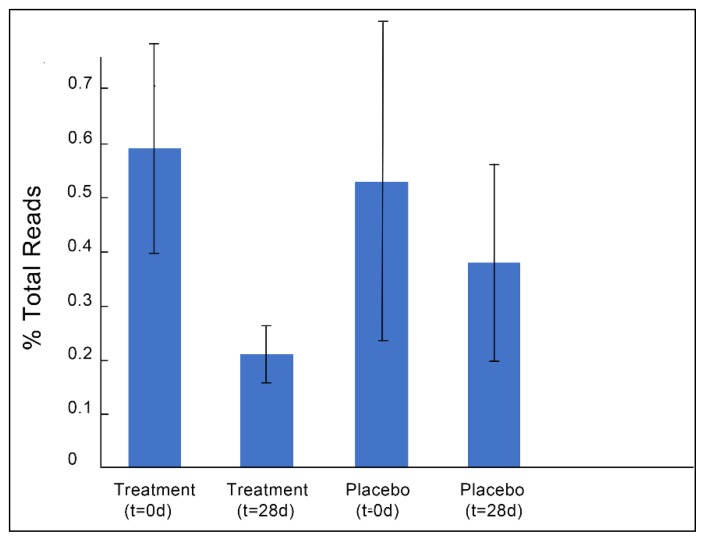
(**A**) Percent of total reads represented by amplicon sequence variants (ASVs) mapping to *Escherichia. coli* for each treatment and time point. (**B**) Change in *E. coli* levels from baseline values after treatment or placebo consumption. Data represents only individuals with baseline *E. coli* levels (*n* = 21). Error bars represent SEM.

**Figure 3 nutrients-11-00666-f003:**
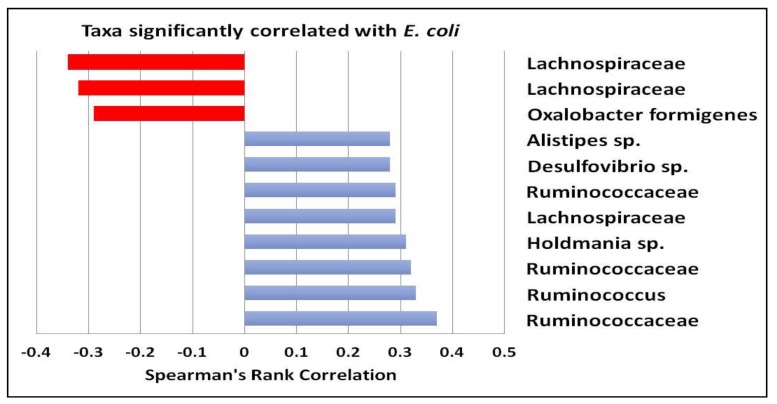
Using Spearman’s rank, several ASVs were found to be significantly negatively correlated (red bars) or positively correlated (blue bars) with *E. coli* ASVs. Significant values were considered *q* < 0.10.

**Figure 4 nutrients-11-00666-f004:**
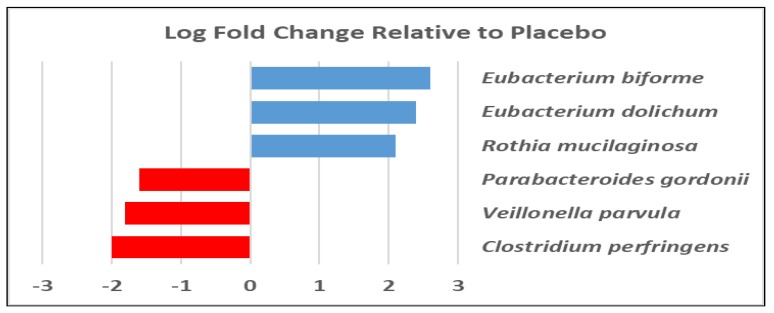
Using a negative binomial generalized linear model (GLM) (EdgeR), we identified several taxa that significantly (*q* < 0.10) differed from placebo levels after 28days of phage consumption. Red bars represent taxa reduced with phage treatment, and blue bars represent taxa that were increased.

**Table 1 nutrients-11-00666-t001:** Stool and plasma lipid profiles.

	Treatment (*t* = 0)	Treatment (*t* = 28)	Placebo (*t* = 0)	Placebo (*t* = 28)
**Fecal Triglycerides (mg/dL)**	6.70 (±0.70)	8.12 (±0.89)	9.25 (±1.14)	7.46 (±0.79)
**Cholesterol (mg/dL)**	189.70 (±4.93)	187.18 (±4.83)	192.03 (±5.76)	189.35 (±6.08)
**LDL (mg/dL)**	103.48 (±3.56)	100.82 (±4.23)	106.76 (±4.85)	103.06 (±4.67)
**vLDL (mg/dL)**	20.21 (±1.82)	20.52 (±1.87)	19.62 (±1.94)	19.85 (±1.94)
**HDL (mg/dL)**	65.06 (±2.51)	65.24 (±2.58)	65.74 (±2.33)	63.91 (±2.84)
**nHDLc (mg/dL)**	123.61 (±4.89)	122.85 (±4.98)	126.21 (±5.68)	125.62 (±5.40)
**TC/H**	3.03 (±0.13) ^a^	3.01 (±0.14) ^a^	3.02 (±0.13) ^a^	3.11 (±0.14) ^b^
**Plasma Triglycerides (ng/dL)**	99.94 (±9.11)	102.61 (±9.46)	97.74 (±9.70)	99.41 (±9.77)

Data represent means (±SD). Statistically different values are denoted with different letters (*p*<0.05). Total cholesterol (TC), high-density lipoprotein cholesterol (HDL), nonhigh-density lipoprotein cholesterol (nHDL), low-density lipoprotein cholesterol (LDL), and very low-density lipoprotein cholesterol (vLDL).

**Table 2 nutrients-11-00666-t002:** Plasma C-reactive protein (CRP) and cytokines.

	Treatment (*t* = 0)	Treatment (*t* = 28)	Placebo (*t* = 0)	Placebo (*t* = 28)
**hsCRP (mg/mL)**	1.76 (±0.51)	1.79 (±0.52)	1.56 (±0.41)	2.45 (±0.68)
**GMCSF (pg/mL)**	80.69 (±9.99)	80.54 (±10.35)	83.17 (±9.78)	80.59 (±10.10)
**IFN-γ (pg/mL)**	12.67 (±1.12)	12.29 (±1.02)	14.21 (±1.86)	13.81 (±1.89)
**Il-10 (pg/mL)**	24.13 (±3.30)	23.53 (±2.85)	26.03 (±3.75)	24.83 (±3.84)
**Il-12 (pg/mL)**	3.57 (±0.33)	3.57 (±0.32)	3.75 (±0.35)	3.55 (±0.37)
**Il-13 (pg/mL)**	23.05 (±5.53)	22.16 (±5.71)	25.39 (±6.06)	24.5 (±5.73)
**Il-1β (pg/mL)**	1.76 (±0.13)	1.71 (±0.10)	1.85 (±0.13)	1.73 (±0.12)
**Il-2 (pg/mL)**	2.24 (±0.14)	2.15 (±0.18)	2.46 (±0.28)	2.36 (±0.31)
**Il-4 (pg/mL)**	69.48 (±5.75) ^a^	59.83 (±4.43) ^b^	63.79 (±4.95) ^a,b^	61.71 (±3.88) ^a,b^
**Il-5 (pg/mL)**	8.16 (±3.26)	5.55 (±1.35)	7.85 (±2.71)	6.38 (±1.57)
**Il-6 (pg/mL)**	3.37 (±0.33)	3.43 (±0.33)	3.76 (±0.38)	3.82 (±0.41)
**Il-7 (pg/mL)**	13.37 (±1.07)	13.39 (±1.14)	13.91 (±1.16)	13.35 (±1.12)
**Il-8 (pg/mL)**	4.14 (±0.70)	4.24 (±0.77)	4.51 (±0.83)	4.45 (±0.81)
**TNFα (pg/mL)**	4.45 (±0.29)	4.18 (±0.25)	4.23 (±0.24)	4.09 (±0.26)

Values represent mean (± SEM). Statistically significant differences are denoted by different letters (*p* < 0.01). Il: Interleukin; GMCSF: Granulocyte Macrophage Colony Stimulating Factor; TNF: Tumor Necrosis Factor.

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
