# Peer review of "PHAGE Study: Effects of Supplemental Bacteriophage Intake on Inflammation and Gut Microbiota in Healthy Adults"

_nutrients, 2019, doi:10.3390/nu11030666_

Reviewer 1 Report

In the study of Febvre et al., the authors aim to evaluate the effect of a E.coli-targeting phages daily supplement (PreforPro) on the gut microbiota composition in healthy subjects. The gut microbiota composition, the SCFAs and some inflammatory parameters have been evaluated in subjects previously enrolled in a double blinded, randomized, placebo-controlled crossover trial with 28-day intervention periods separated with a 2 weeks washout period, implemented by the same authors to test the safety of the supplemental bacteriophage consumption. The authors affirm that treatment did not provoke a global perturbation in the gut microbiota composition, documenting only minimal alterations that may be viewed as beneficial, with the reduction of potentially pro-inflammatory bacteria. Finally, the authors suggest the use of bacteriophages as potentially therapeutic agents in clinical populations of intestinal inflammation, underlying the benefit of selectively modifying gut microbial community without affecting the whole gut microbiota (as it happen with the use of antibiotics).

The study is really interesting and original. Anyway, it needs major and minor revisions. In particular, I strongly recommend to be more specific in the description of results reporting both in the text and in tables (or figures legend) the results of statistical analysis (i.e. p values). Moreover, the symbols (i.e. Pag. 4 line 143-144) are missing throughout the text.

Introduction, page 2 line 60: “phages offer the opportunity to subtly …modify the gut microbiota” : please add reference.

Introduction page 2 line 49: can authors add some example of studies regarding the use of bacteriophages as microbial modifiers in clinical practice or in clinical research?

Introduction, page 2 line 45: “according to transparency market research…by 2022”: please add reference

Material and Methods, page 3 line 102: please add a reference.

Material and Methods, page 3 line106 cut the comma after “Invitrogen;”

Material and Methods, page 3 lines 116-117, please add detail about generation of OTUs table

Material and Methods, page 3 line 120: lack the accession number

Material and Methods, page 3 lines 122-128: Please describe the specific SCFAs that will be detect, the GC instrument features and the temperature programs.

Material and Methods, page 4 lines 143-144: add symbols (IFN-alfa, IL-1 beta ecc..)

Material and Methods, page 4 Statistical Analysis: I suggest assessing the beta diversity also with the Weighted UniFrac dissimilarity and perform the PCoA visualization of Weighted UniFrac distances. Moreover, I suggest to perform PERMANOVA analysis to compare the entire bacterial composition between treatments and time points.

Material and Methods, page 4 line 167: please change Kruskal-Wallace with Kruskal-Wallis

Results page 4 line 175: Correct Verrucomicrobiota with Verrucomicrobia.

Results page 4 line 175: The text report no significant differences in bacterial taxa between treatments and time points. Anyway, Figure 1A reports only differences in relative abundances of phyla detected in stool samples. I suggest to add bar plots of relative abundances at each taxonomic level at least as supplementary material.

Figure 1B please add the % of variance to the axis 1 and 2

Tables S1, S2, Figure S1 and S2: please add the results of the statistical analysis and provide a detail description of the table/figure under the caption with a complete description of abbreviations.

Results page 5 line 191: please add the p value result for the placebo.

Figure 2 A: I suggest to use the box plot representation instead of bar plot, reporting median +/- IQR instead of SEM.

Results page 5 line 200: correct Desulfovibrio sp. With Desulfovibrio spp.

Results page 6 lines 208-210. The result regarding P. gordonii is not clear, please examine the sentence and add the comparison between Treatment t=0 and Treatment=28 for P. gordonii in Table S2. In particular, it seems the authors want to say that the reduction of P.gordonii after treatment period is not the consequence of the treatment but is due to its increase during placebo period. Anyway, in the cross over design, treatment can precede or follow the treatment period, so I don’t understand what the authors state.

Tables S2: please for completeness, add comparison between placebo timepoints also for B. bifidum

Results page 6 line 214: please add reference about C. perfringens

Figure 3. Please correct Ruminocaceae with Ruminococcaceae

Figure 4 legend: The figure report the taxa differences from placebo or from baseline levels??. Moreover, please add bar description.

Results page 7 line 222: correct proprionate with propionate.

Results page 7 lines 224-225: the authors say that there is a “trend towards increased acetate from baseline levels within the placebo period”. Anyway the figure S1A shows the contrary.

Results page 7 line 227: “ since the microbiota also play a role in lipid adsorption” please add a reference.

Table 1 and Table 2: What the values represent? Mean +/- SEM? Please add a description and show the results and of statistical analysis (i.e. p values).  

Results page 7 line 240: immunological instead of immunologic

Discussion page 8 line 271 and 281: Eubacterium spp. And Desulfovibrio spp.

Discussion page 9 lines 289-291. This sentence should be cut as it has no valid scientific foundation.

Also I suggest to report in the supplementary material a table with Microbial taxa variations (abundances: median and interquartile range, IQR) between treatments and timepoints calculated using Kruskal-Wallis test.

Finally I can find any results or comments about the PICRUSt analysis. Why? The authors should write something about since they performed it.

Author Response

We would like to thanks the reviewers for their careful assessment and helpful comments with the manuscript. Please find our responses below to specific issues raised by the review process.

Reviewer 1 mentioned several places where references should be added or where typographic/grammatical errors required correction. In addition, reviewer 1 requested that the appropriate symbols be added for TNF-alpha, IL-1beta, etc.

Thank you for the careful reading of our manuscript. Where appropriate, we have added references as requested by the reviewer and corrected the noted errors in language and spelling. Specific changes can be noted in the highlighted version of the manuscript. Regarding the symbols, in the authors PDF version the appropriate symbols are visible in the indicated places. However, we will continue to work with copyeditors to ensure the appropriate use and placement of symbols if the manuscript continues towards publication.

Reviewer 1 requested information regarding generation of OTU table.

Thank you for bringing up this point. We have incorrectly used the terminology OTU throughout the manuscript. Our initial sequence processing method was DADA2, which does not use the OTU clustering methodology, but rather amplicon sequence variant (ASV) assignment which is based on algorithms detecting true biological variants prior to the introduction of sequencing errors. This method both reduces artifacts and improves resolution between variants and has been widely adopted as a default processing method in many bioinformatics pipelines. We have clarified this throughout the manuscript and included a description of ASV generation in the Methods section.

Reviewer 1 recommended the use of Weighted Unifrac distances rather than Bray-Curtis for B-diversity visualization.

While both of these distance/dissimilarity measures are widely used, we prefer to use the Bray-Curtis distance as it takes number and abundance of variants into account (similar to weighted Unifrac), but does not make the assumptions of phylogenetic relatedness that are inherent in the Unifrac calculation. We typically do not incorporate this measure in our analysis; however, it is likely that the results would not be dramatically altered using this distance measure since there was not even a trend of treatment- related clustering of samples in the current analysis. This lack of global effects on the microbiota is unsurprising given 1) phages are highly host specific to their target bacteria, 2) there was very minimal E. coli (phage hosts) within our study population; and 3) given this was a longitudinal study, the interpersonal differences are unlikely to be overcome by a treatment meant to elicit only very subtle effects. Thus, we have retained the Bray-Curtis visualization and the perMANOVA analysis is in the figure legend.

Reviewer 1 has asked that bar plots at other taxa levels (besides Phyla) are added to the manuscript as supplemental material.

We have generated these plots and added them to the supplemental data as Supplemental Figure 1.

Reviewer 1 has asked that we add the % of variance on the axes of Fig 1B.

Figure 1B is a plot of the Bray-Curtis distances visualized by Principle Coordinates Analysis (PCoA-specifically non-metric dimensional scaling), thus there are no percent variances associated with the axes. This type of plot does generate a stress value and we have reported this in the figure legend.

Reviewers 1 and 3 have asked for more complete reporting of the statistical values in tables and figures (including supplemental).

We have include p-values and other appropriate statistical parameters where requested.

Reviewer 1 enquired why box and whisker plots were not used to show differences across groups in E. coli levels.

We did generate these plots, but their usefulness in the ability to visualize differences between groups was limited due to the high level of variability in E. coli populations between participants (ranged from 0.001-3.5% of the total reads). The boxes were very compressed near the x axis because sample means were small, but there were a couple individual datapoints that were very high. Therefore, we have left this figure as it was.

There was some confusion regarding our statements regarding changes in P. gordonii during the treatment period.

We agree that the discussion around this is confusing and ultimately irrelevant to the conclusions of the study (P. gordonii varied significantly across nearly all timepoints/treatments suggesting it may be extremely susceptible in general to perturbations or has cyclic population growth and decline but really wasn’t toed to any specific treatment or placebo effect. As a result we have removed this section and the corresponding supplemental table and placed remaining information from the table that was relevant directly into the text.

Reviewer 1 has requested requested inclusion of a statement about the PiCRUST analysis, and to generate tables with microbial taxa variations between timepoints with Kruskal Wallis test results.

We have removed reference to  PiCRUST analysis as we saw nothing of significance to report in the paper. With regards to the requested table, we did not find any significant differences using an ANCOVA model. The specific taxa shown in our log fold change graphs represent differential abundance between groups/time based on an EdgeR analysis (a negative binomial GLM) that models microbiota count data using an over-dispersed Poisson model, which includes an empirical Bayes procedure to moderate the degree of over-dispersion typical of 16s data. Thus, EdgeR overcomes some of the standard statistical limitations encountered when dealing with highly dispersed and sparse datasets like 16s sequences to allow detection of differentially abundant taxa.

Reviewer 3 requested a more detailed description of the treatment provided.

We have added more detail regarding the conduct of the human study, including the clarification of the participant numbers and a description of the treatments provided. We have also clarified the instructions that were given to participants regarding storage of the stool samples in the time between collection and storage in the lab.

Reviewer 3 also commented on both the low abundance of E. coli as well as the high variability within our study population and suggested that a population with baseline higher E. coli would have been desirable.

We agree that it is difficult to discern the effects of a cocktail of E. coli-containing phages in a population with minimal E.coli. However, the intent of our study was two-fold. 1) to test this phage cocktail in a generally healthy population as it is commercially marketed as a dietary supplement for GI health and 2) establish the safety and tolerability of phage consumption in a non-diseased  population in to open the door for future studies in which this approach might be studied in a population with high baseline E. coli (such as individuals with Crohn’s disease). Thus, this was what we hope to be an initial step in future phage research. We have elaborated on this somewhat in the discussion.

 Reviewer 3 requested clarification regarding the terms “responder” and “larger population”.

We have clarified this in the manuscript by directly defining these parameters. The “responder” population are individuals that had baseline levels of E. coli which were reduced by phage consumption (n=15), while the large population referred to microbiota changes in the entire study population (n=36).

 Reviewer 3 commented on the lack of diet and physical activity data as potential confounding factors in the experimental design.

We agree with the reviewer that this data would have strengthened the study. We did instruct participants to maintain their standard diet and exercise regime and we refrained from having volunteers participate during holidays and other times when diet deviations were more likely to occur to minimize these effects. We also acknowledge that this is one of the major weaknesses of the study. We have elaborated on this in the discussion. The reviewer may be interested to note that we are conducting a follow-up study where these parameters are better monitored.

Reviewer 2 Report

Dear Authors,

The study is very well designed and executed very well. Bacteriophage, microbiota and IBD have increasingly linked. Analysis with plasma lipid status and cytokines without significant change showed participant similarity (may be food intake and life style). Further studies may need at least  control diet (with low fiber or high fiber). The novel finding is reduction in Clostridium and increase SCFAs producing Eubacterium. Further studies with experimental model to test the bacteriophage effect on Clostridium is important to reveal inverse correlation between Bacteriophages and Clostridium.

Author Response

(The authors gave the same response as above.)

Reviewer 3 Report

This manuscript is the continuation of a previous work (reference 7) where the authors tested the safety and tolerability of the phage preparation. In this study, authors present the results from the analysis of fecal microbiota and blood samples taken from the participants of the previous study.

The first issue I see is that methods section is missing the most important part, the description of the treatment in detail. Authors do mention the reference to the previous work but I think that in this study they should also include a description of treatment “A” and treatment “B” including the composition of the phages that they have used. In addition, authors mention that 43 participants were enrolled in the study but from these only 32 completed the whole study and therefore this should also be mentioned in the study design. Moreover, regarding stool samples collections, it would be best if they also specify in which exact conditions were kept from collection to reception of these samples. Were they kept at room temperature or refrigerated/frozen?

Regarding figure 2, the percentage of total reads is very low (0.6% approx. in the highest groups) and the error bars are quite big as well. This is actually expected as they are healthy participants and, therefore, no significant levels of E. coli is expected. Indeed, several participants didn’t even show baseline E coli levels at all. In this sense, the study design I think is not the best as authors are using phages targeting E coli and therefore, it would have been better to target other type of participants.  

Authors state that Bifidobacterium bifidum was the only OTU that increased significantly after the phages treatment in the responders population while in figure 4 they show the OTUs that were significantly changed after treatment in the larger participant population. Please, clarify “responders” and “larger participant population”, I feel is not clear in the text what they mean with that participants differentiation. Again, this may be explained in more detail in the first manuscript but it should also be explained here.

Diet and physical activity assessment is missing as the authors actually mention in discussion and this should have been taken in consideration as its influence on microbiota could be huge.

As a small note, figures format can be improved significantly. Moreover, stats should be added in tables (ie IL4).

Overall, I think that the use of bacteriophages to modulate gut microbiota is a novel approach that needs to be further investigated but the present study does not report significant differences except for reductions in IL4 and some of the bacteria taxa.

Author Response

We would like to thanks the reviewers for their careful assessment and helpful comments with the manuscript. Please find our responses below to specific issues raised by the review process.

Reviewer 1 mentioned several places where references should be added or where typographic/grammatical errors required correction. In addition, reviewer 1 requested that the appropriate symbols be added for TNF-alpha, IL-1beta, etc.

Thank you for the careful reading of our manuscript. Where appropriate, we have added references as requested by the reviewer and corrected the noted errors in language and spelling. Specific changes can be noted in the highlighted version of the manuscript. Regarding the symbols, in the authors PDF version the appropriate symbols are visible in the indicated places. However, we will continue to work with copyeditors to ensure the appropriate use and placement of symbols if the manuscript continues towards publication.

Reviewer 1 requested information regarding generation of OTU table.

Thank you for bringing up this point. We have incorrectly used the terminology OTU throughout the manuscript. Our initial sequence processing method was DADA2, which does not use the OTU clustering methodology, but rather amplicon sequence variant (ASV) assignment which is based on algorithms detecting true biological variants prior to the introduction of sequencing errors. This method both reduces artifacts and improves resolution between variants and has been widely adopted as a default processing method in many bioinformatics pipelines. We have clarified this throughout the manuscript and included a description of ASV generation in the Methods section.

Reviewer 1 recommended the use of Weighted Unifrac distances rather than Bray-Curtis for B-diversity visualization.

While both of these distance/dissimilarity measures are widely used, we prefer to use the Bray-Curtis distance as it takes number and abundance of variants into account (similar to weighted Unifrac), but does not make the assumptions of phylogenetic relatedness that are inherent in the Unifrac calculation. We typically do not incorporate this measure in our analysis; however, it is likely that the results would not be dramatically altered using this distance measure since there was not even a trend of treatment- related clustering of samples in the current analysis. This lack of global effects on the microbiota is unsurprising given 1) phages are highly host specific to their target bacteria, 2) there was very minimal E. coli (phage hosts) within our study population; and 3) given this was a longitudinal study, the interpersonal differences are unlikely to be overcome by a treatment meant to elicit only very subtle effects. Thus, we have retained the Bray-Curtis visualization and the perMANOVA analysis is in the figure legend.

Reviewer 1 has asked that bar plots at other taxa levels (besides Phyla) are added to the manuscript as supplemental material.

We have generated these plots and added them to the supplemental data as Supplemental Figure 1.

Reviewer 1 has asked that we add the % of variance on the axes of Fig 1B.

Figure 1B is a plot of the Bray-Curtis distances visualized by Principle Coordinates Analysis (PCoA-specifically non-metric dimensional scaling), thus there are no percent variances associated with the axes. This type of plot does generate a stress value and we have reported this in the figure legend.

Reviewers 1 and 3 have asked for more complete reporting of the statistical values in tables and figures (including supplemental).

We have include p-values and other appropriate statistical parameters where requested.

Reviewer 1 enquired why box and whisker plots were not used to show differences across groups in E. coli levels.

We did generate these plots, but their usefulness in the ability to visualize differences between groups was limited due to the high level of variability in E. coli populations between participants (ranged from 0.001-3.5% of the total reads). The boxes were very compressed near the x axis because sample means were small, but there were a couple individual datapoints that were very high. Therefore, we have left this figure as it was.

There was some confusion regarding our statements regarding changes in P. gordonii during the treatment period.

We agree that the discussion around this is confusing and ultimately irrelevant to the conclusions of the study (P. gordonii varied significantly across nearly all timepoints/treatments suggesting it may be extremely susceptible in general to perturbations or has cyclic population growth and decline but really wasn’t toed to any specific treatment or placebo effect. As a result we have removed this section and the corresponding supplemental table and placed remaining information from the table that was relevant directly into the text.

Reviewer 1 has requested requested inclusion of a statement about the PiCRUST analysis, and to generate tables with microbial taxa variations between timepoints with Kruskal Wallis test results.

We have removed reference to  PiCRUST analysis as we saw nothing of significance to report in the paper. With regards to the requested table, we did not find any significant differences using an ANCOVA model. The specific taxa shown in our log fold change graphs represent differential abundance between groups/time based on an EdgeR analysis (a negative binomial GLM) that models microbiota count data using an over-dispersed Poisson model, which includes an empirical Bayes procedure to moderate the degree of over-dispersion typical of 16s data. Thus, EdgeR overcomes some of the standard statistical limitations encountered when dealing with highly dispersed and sparse datasets like 16s sequences to allow detection of differentially abundant taxa.

Reviewer 3 requested a more detailed description of the treatment provided.

We have added more detail regarding the conduct of the human study, including the clarification of the participant numbers and a description of the treatments provided. We have also clarified the instructions that were given to participants regarding storage of the stool samples in the time between collection and storage in the lab.

Reviewer 3 also commented on both the low abundance of E. coli as well as the high variability within our study population and suggested that a population with baseline higher E. coli would have been desirable.

We agree that it is difficult to discern the effects of a cocktail of E. coli-containing phages in a population with minimal E.coli. However, the intent of our study was two-fold. 1) to test this phage cocktail in a generally healthy population as it is commercially marketed as a dietary supplement for GI health and 2) establish the safety and tolerability of phage consumption in a non-diseased  population in to open the door for future studies in which this approach might be studied in a population with high baseline E. coli (such as individuals with Crohn’s disease). Thus, this was what we hope to be an initial step in future phage research. We have elaborated on this somewhat in the discussion.

Reviewer 3 requested clarification regarding the terms “responder” and “larger population”.

We have clarified this in the manuscript by directly defining these parameters. The “responder” population are individuals that had baseline levels of E. coli which were reduced by phage consumption (n=15), while the large population referred to microbiota changes in the entire study population (n=36).

 Reviewer 3 commented on the lack of diet and physical activity data as potential confounding factors in the experimental design.

We agree with the reviewer that this data would have strengthened the study. We did instruct participants to maintain their standard diet and exercise regime and we refrained from having volunteers participate during holidays and other times when diet deviations were more likely to occur to minimize these effects. We also acknowledge that this is one of the major weaknesses of the study. We have elaborated on this in the discussion. The reviewer may be interested to note that we are conducting a follow-up study where these parameters are better monitored.

Round  2

Reviewer 1 Report

I want to thank the authors for having revised the manuscript in accordance with my suggestions and for having provided extensive responses to my comments. I accept the pubblication in the present format.

Author Response

Thank you for the thoughtful review.

Reviewer 3 Report

I am happy with authors responses. 

Figure 2 could be improved in terms of format (maybe using other software different to excel such as graphpad). 

In table 1, I dont think that there is significant references for the TC/H, it seems very tight considering the error observed (ie, diference between 3 and 3.1 cannot be sifnificant).

Author Response

Thank you for your thoughtful review. In accordance with your suggestion, we have re-made figure 2A using graph pad. We removed Figure 2B as it added little to the overall analysis and instead we stated the % reduction in reads in the text. 

We rechecked the values and re-ran the model for TC/HDL and it does come out to be statistically significant. However, we, agree that these actual values have no clinical or biological relevance so we stated this in the text. 

Line 265:

However, these statistical differences are unlikely to have any clinical or biological relevance.